# Thermodynamics of Vacuum Chloride Volatilization of Ni, Co, Mn, Li, Al, and Cu in Spent Lithium−Ion Battery

**Wen Luo [1,2], Guochen Hu [1,2], Junshuai Ding [1], Jijun Wu [1,2,\*] and Wenhui Ma [1,2]**

1    Faculty of Metallurgical and Energy Engineering, Kunming University of Science and Technology, Kunming 650093, China
2    National Engineering Laboratory of Vacuum Metallurgy, Kunming University of Science and Technology, Kunming 650093, China
\*    Correspondence: dragon_wu213@126.com; Tel.: +86-13888411653

**Abstract:** In recent years the chlorination leaching separation process in the field of hydrometallurgy has been developed considerably. However, the development of the chlorination separation process in the field of pyrometallurgy has lagged. In this paper, the thermodynamics of vacuum chlorination volatilization of valuable metals Ni, Co, Mn, Li, Al, and Cu from spent lithium−ion batteries is investigated, and it is found that chlorination helps to achieve the trapping and separation of the singlet metals. With the help of Factsage 8.1, a theoretical map of the stable regions of valuable metal chlorides, the order of separation of each chloride at 10 Pa, and a discussion of the behavior of excess $CuCl_2$ in the system at different temperatures were determined. This paper provides research ideas in the fields of selective separation of alloying elements in the carbothermal reduction products of waste lithium−ion batteries and one−step separation of valuable metals by carbothermal reduction.

**Keywords:** spent lithium−ion battery; thermodynamics; chloride volatilization; phase diagram analysis

## 1. Introduction

With the shortage of metal resources such as Ni, Co, and Li, the prices of these valuable metals continue to soar. In order to meet the resource requirements for the production of lithium−ion batteries, recycling spent batteries is going to be of increasing importance. The mainstream method for the extraction of valuable metals from waste lithium−ion batteries (LIBs) is to use inorganic acids [1], organic acids [2] or mixed acids [3] to transfer the valuable metals into the solution system, and then use oxalate [4], carbonate [5] and other precipitants to gradually separate the valuable metals Ni, Co, Mn, and Li by adjusting the potential of hydrogen (pH). In this process, nearly pure valuable metal compounds can be recovered [6]. Acid leaching plays an indispensable role in removing valuable metals from spent LIBs, but the problem of treating waste acid and exhaust gas has always impeded its further application.

As a viable solid waste treatment method, the chlorination process is widely used in the treatment and recovery of heavy metals from municipal solid waste [7,8], tailings [9], medical waste [10] and nuclear waste [11]. Hydrochloric acid was gradually replaced by non−volatile chlorinating agents such as $CaCl_2$ [12], NaCl [13], and $MgCl_2$ [14] in order to prevent acid loss and air pollution caused by its volatilization.

According to relevant research [15], nearly half of the studies in the field of lithium−ion battery recycling which use acid leaching technology still use hydrochloric acid as a reducing agent or leaching agent. In order to improve this situation, some studies [16] proposed chlorinating the valuable metal Li in the waste battery using the chlorinating agent $CaCl_2$. This was done by immersing it in water to obtain an aqueous solution of LiCl for subsequent treatment. This process not only separates the valuable metal Li from the lithium battery waste by strictly controlling the conditions, but also has superior selectivity. However, the

chlorinating agent used is easily soluble in water and will introduce additional impurity elements; and the obtained by−products will react with water to complicate the solution system, which has brought significant impact and challenges to the ecological environment. Research using the chlorination process is mainly carried out under normal pressure and low temperature [17–19]. Therefore, it is particularly pertinent to study the thermodynamics of chloride transformation and volatilization under the coupled vacuum−high temperature condition.

The thermodynamics of vacuum chloride volatilization of Ni, Co, Mn, Li, Al, and Cu in spent lithium−ion battery [15] cathode materials were investigated in this paper. Firstly, starting from the elements contained in the waste lithium battery, through thermodynamic calculation, the appropriate chlorinating agent is selected without introducing new impurities. Analyze the thermodynamic equilibrium diagram of the chloride system of each element and determine the possibility and conditions of the reaction from the perspective of theoretical calculation. The method for separating valuable metals by vacuum chlorination, based on pyrometallurgy, solves the problem that these elements are difficult to separate and purify due to their similar physical and chemical properties, and provides a solution for the subsequent treatment of related wastes.

## 2. Thermodynamic Calculation

The thermodynamic data in this paper were obtained and computed by Factsage 8.1. Pressure has a significant influence on the boiling point (sublimation point) of a substance. Generally, the isothermal equation of Van 't Hoff chemical reaction is used. The Gibbs free energy change of the reaction under actual conditions is calculated and deduced. The boiling point of the substance under the corresponding conditions is indirectly determined [20]. Use Equation (2) to calculate the $\Delta G$ of the general chemical reaction equation (Equation (1)) and apply the interpolation method to obtain the corresponding temperature when $\Delta G = 0$, which is the boiling point of the substance under the corresponding pressure.

$$\text{Compound(s, l)} = \text{Compound(g)} \tag{1}$$

$$\Delta G = \Delta G^{\theta} + \frac{\left[ 2.303 \mathrm{R} T \lg \left( \frac{\mathrm{p}_{Cl_2}}{\mathrm{p}^{\theta}} \right) \right]}{1000} + \sum_{i} \left[ \Delta H_{phi} - \Delta H_{phi} \left( \frac{T}{T_{phi}} \right) \right] \tag{2}$$

In the equation, $\Delta G$ is the Gibbs free energy change of the reaction under actual conditions; $\Delta G^{\theta}$ is the Gibbs free energy change of the reaction under the standard condition; R = 8.314 J/(mol·K), is the gas constant; T is the actual temperature; $p_{Cl_2}$ is the partial pressure of $Cl_2$ in the system; $p^{\theta}$ = 101,325 Pa, is the pressure under standard conditions; $\Delta H_{phi}$ is the enthalpy change of the phase transition of the substance; $T_{phi}$ is the temperature at which the substance undergoes phase transition, the unit is K. As shown in Table 1, the melting and boiling points (sublimation points) of the corresponding chlorides of Ni, Co, Mn, Li, Al, and Cu are listed, respectively, at 101,325 Pa and 10 Pa. The boiling points of all the chlorides listed, except LiCl and CuCl, are reduced below their melting points at 10 Pa. This property helps to reduce the interaction of the chlorides in the molten state and facilitates the separation between them.

**Table 1.** Melting and boiling points (sublimation points) of Ni, Co, Mn, Li, Al, and Cu corresponding chlorides at 101,325 Pa and 10 Pa.

| Chloride | Melting Point (K) | Boiling Point (K) | |
|---|---|---|---|
| | | 101,325 Pa | 10 Pa |
| LiCl | 883.00 | 1701.55 | 996.78 |
| NiCl$_2$ | 1304.00 | 1242.79 * | 866.15 * |
| CoCl$_2$ | 1010.00 | 1371.13 | 852.13 * |
| MnCl$_2$ | 923.00 | 1510.24 | 888.29 * |
| MnCl$_3$ | 860.00 | 953.96 | 630.15 * |
| AlCl$_3$ | 465.70 | 623.74 | 399.46 * |
| CuCl | 696.00 | 1950.80 | 1094.66 |
| CuCl$_2$ | 871.00 | 1094.04 | 726.62 * |

* Sublimation Point.

In order to avoid the introduction of new impurity elements, it is necessary to select suitable chlorides as the chlorinating agent in the valuable metals of the waste ternary lithium battery system. The general equation of the chlorination reaction can be described as follows:

$$\frac{2}{x}Me + Cl_2(g) = MeCl_x \ (Me = Li, Ni, Co, Mn, Al, Cu; x = 1, 2, 3) \qquad (3)$$

Using the classical thermodynamic method, the stoichiometric number 1 of Cl$_2$(g) in the reaction equation was used as the standard to calculate the $\Delta G-T$ relationship between Li, Ni, Co, Mn, Al, Cu, and other metal elements reacting with Cl$_2$(g). As shown in Figure 1, graph (a) lists the $\Delta G^\theta-T$ relationship of each reaction under 101,325 Pa, and graph (b) lists the $\Delta G-T$ relationship under the condition of 10 Pa. Figure 1 shows the phase transition point (T), melting point (M), boiling point (B) and sublimation point (S) of each substance under different gas pressures. The boiling point of the substance changes with the pressure in the system: the decrease in pressure will greatly affect the boiling point of the substance [21]. Under 101,325 Pa, only NiCl$_2$ sublimates at 1242.79 K; under 10 Pa, NiCl$_2$ sublimates at 866.15 K; other substances that do not sublime under 101,325 Pa, such as CoCl$_2$, Mn, MnCl$_2$, MnCl$_3$, CuCl$_2$, etc., all have the sublimation phenomenon when the system pressure is 10 Pa. The property that the boiling point (sublimation point) of the substance decreases with the decrease in the gas pressure provides the possibility of realizing the subsequent separation and purification of chlorides in a vacuum environment.

From Figure 1a, in the listed temperature range, CuCl$_2$ can chlorinate almost all the metal elements in the system, and AlCl$_3$ can also chlorinate other metals with the increase in the reaction temperature [22,23]. However, considering that the subsequent reaction is carried out at a system pressure of 10 Pa, referring to Figure 1b, it is found that AlCl$_3$ sublimates at 399.46 K, and is then evacuated from the reaction system by the vacuum system, so it cannot be used as chlorine chemical agent [24]. Since the Gibbs free energy of generating AlCl$_3$ is small, and it is prone to sublimation under vacuum, this feature can be used to separate Al in the system [25]. In Figure 1b, the $\Delta G-T$ curves of CuCl$_2$, NiCl$_2$, CoCl$_2$, and MnCl$_2$ are almost parallel when the temperature is in the range of 298–900 K. It shows that in this interval, the temperature has a negligible effect on the thermodynamics of the chlorination reaction. However, under the condition of 10 Pa, CuCl$_2$ decomposes at 644.79 K. In order to avoid its decomposition, the chlorination reaction temperature should be lower than this value.

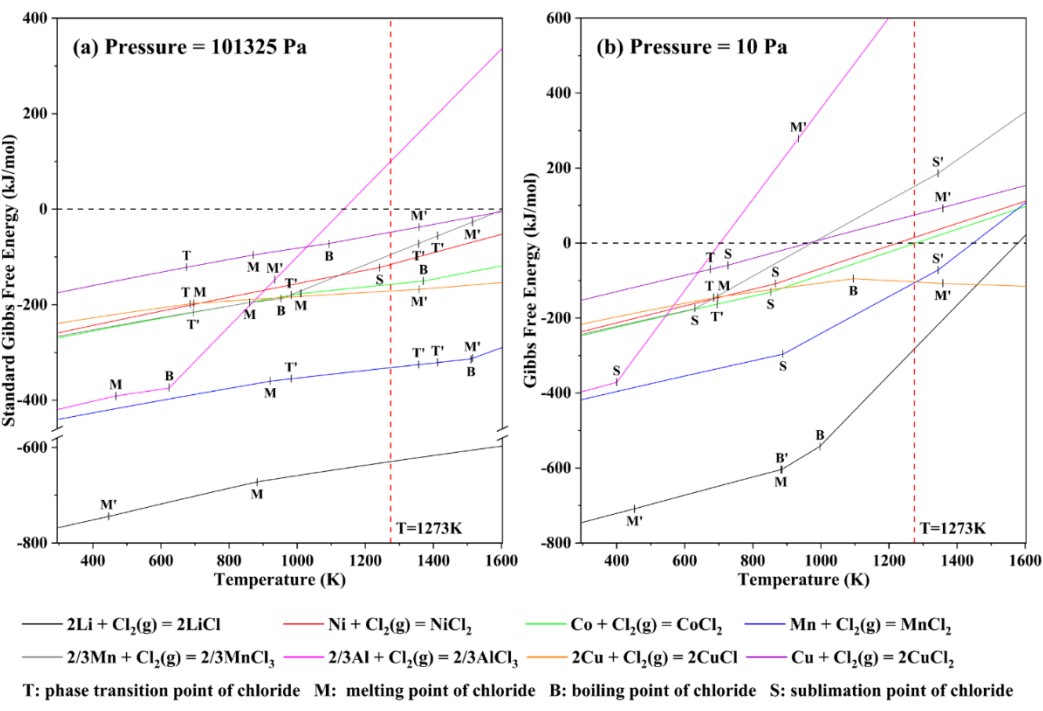

**Figure 1.** ΔG−T relationship between Li, Ni, Co, Mn, Al, Cu, and other metal elements reacted with Cl₂(g): (**a**) 101325 Pa; (**b**) 10 Pa.

## 3. Results and discussion

### 3.1. Analysis of Dominant Area of Ni−Co−Mn−Li−Cl System

Firstly, the decomposition and volatilization of chlorides of the main valuable metals Ni, Co, Mn, and Li contained in waste lithium batteries were analyzed. According to Equation (2), the equilibrium diagrams of Ni−Co−Mn−Li chlorides at 10 Pa, 100 Pa and 101,325 Pa were drawn accordingly. As shown in Figure 2a–d are the equilibrium diagrams of the decomposition reactions of Ni−Cl, Co−Cl, Mn−Cl and Li−Cl systems, respectively. The red, blue, and black lines are the equilibrium lines of the chloride decomposition reaction at 10 Pa, 100 Pa and 101,325 Pa, respectively. Additionally, they are the phase transition divisions of the corresponding components. In the area above the curve, at any temperature, the actual chlorine partial pressure is higher than the equilibrium chlorine partial pressure at the same temperature on the equilibrium line, so the area above the equilibrium line is the stable area of chloride, and the bottom is the stable area of the corresponding metal. The green slashed area in the figure is the actual control area of the experiment [26]. The upper and lower limits of this area correspond to pressures of 100 Pa and 10 Pa, respectively. The boundary on the left is the temperature at which chloride decomposes to produce chlorine gas at the corresponding pressure. The boundary on the right is the temperature at which chloride sublimates or vaporizes at the corresponding pressure. In this range, chloride can be collected as a gas without decomposing Cl₂ and the related metal, which is kind to both experimental equipment and the environment.

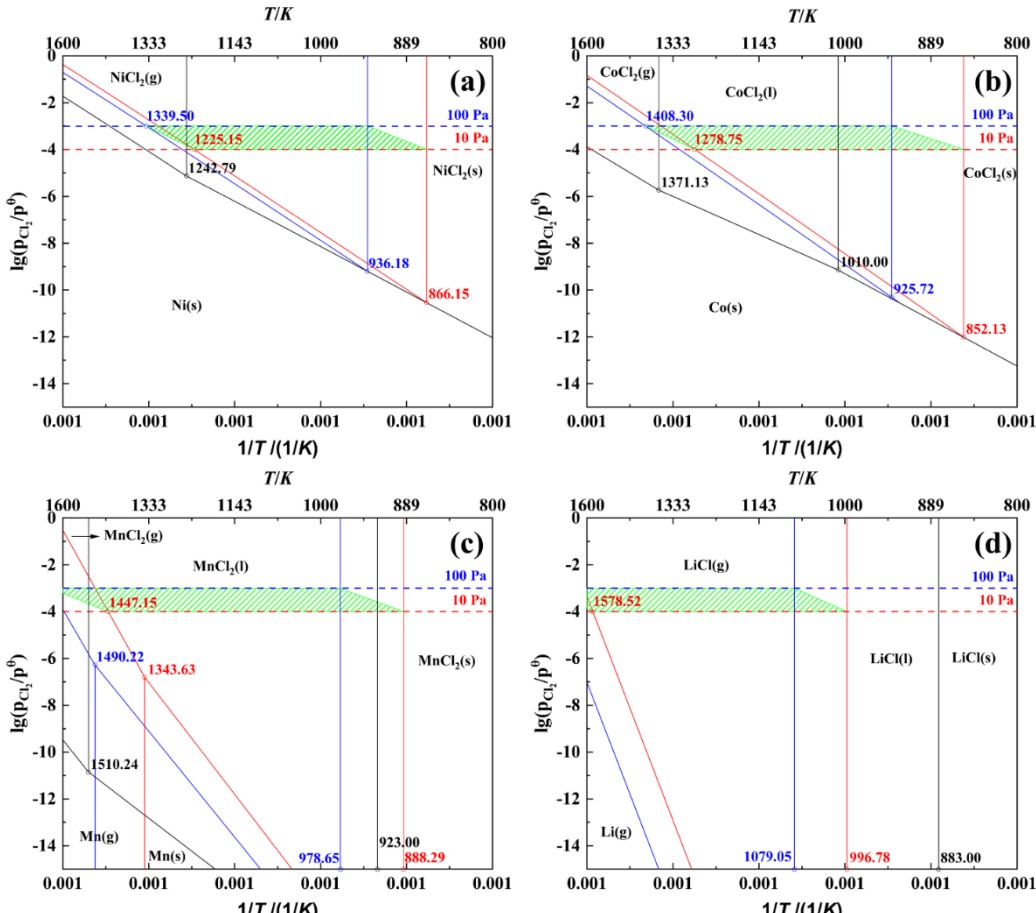

**Figure 2.** Equilibrium diagram of chloride decomposition reaction: (**a**) Ni−Cl system; (**b**) Co−Cl system; (**c**) Mn−Cl system; (**d**) Li−Cl system.

In the equilibrium diagram, the stable zone is directly related to the phase transition temperature: the left and right sides of the sublimation point are the equilibrium zones of the gas and solid phases of the substance; the melting and boiling point, the equilibrium line and the coordinate axis form the stable zone of the liquid phase. And the left and right sides of the liquid phase stable zone are the equilibrium regions of the gas phase and the solid phase, respectively. According to thermodynamic calculations, $NiCl_2$ sublimes in the studied pressure range; $CoCl_2$ sublimes at 10 Pa and 100 Pa; $MnCl_2$ sublimes only at 10 Pa; and LiCl does not sublimes. The introduction of vacuum will reduce the gasification temperature of the material. Under 101,325 Pa, $NiCl_2$, $CoCl_2$ and $MnCl_2$ enter the gas phase at 1242.79 K, 1371.13 K and 1510.24 K, respectively. They drop to 866.15 K, 852.13 K and 888.29 K at 10 Pa. At the same time, the vacuum reduces the decomposition temperature of chlorides. In the studied temperature range, the decomposition temperatures of $NiCl_2$ and $CoCl_2$ decreased from 1339.50 K and 1408.30 K at 100 Pa to 1225.15 K and 1278.75 K at 10 Pa. Moreover, the decomposition temperature of $MnCl_2$ and LiCl at 100 Pa is beyond the research range. The decomposition occurs at 1447.15 K and 1578.52 K at 10 Pa, respectively.

Combining the green areas in Figure 2a–d into the same figure, we can obtain the control regions where Ni−Co−Mn−Li chlorides are separated one by one in gaseous form. As shown in Figure 3, the corresponding chlorides can be obtained by controlling the reaction conditions in the respective intervals from right to left.

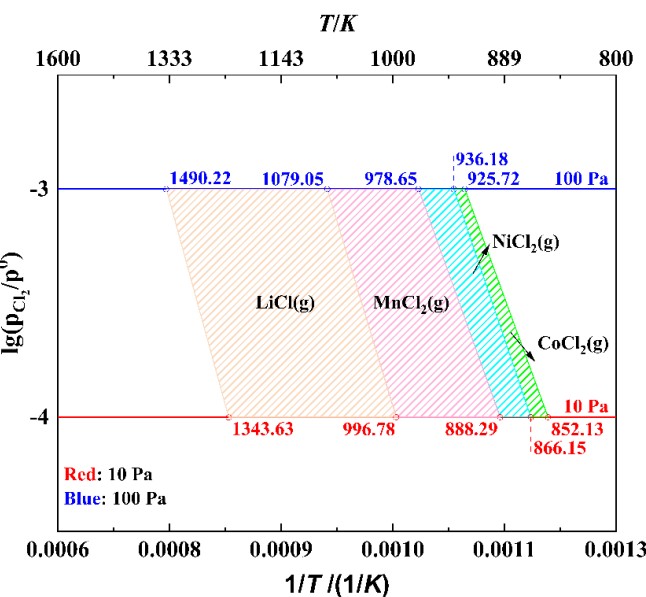

**Figure 3.** Ni−Co−Mn−Li chloride gaseous separation control area.

### 3.2. Analysis of Dominant Area of Al−Cu−Cl System

Figure 4a,b are the equilibrium diagrams of the decomposition reactions of Al−Cl and Cu−Cl chlorides, respectively. The red, blue, and black lines in the figure represent the equilibrium line and the phase transition division of the corresponding components when the system pressure is 10 Pa, 100 Pa and 101,325 Pa, correspondingly. The green slashed area is the actual control area of the experiment. $AlCl_3$ decomposes into Al and $Cl_2$ at 1136.91 K; when the system pressure is 100 Pa, the decomposition temperature of $AlCl_3$ is reduced to 758.82 K; when the system pressure is reduced to 10 Pa, the decomposition temperature of $AlCl_3$ is further reduced to 703.74 K. Under 101,325 Pa, the melting points of CuCl, $CuCl_2$ and Cu are 696.00 K, 871.00 K and 1357.77 K, respectively; at 100 Pa, CuCl and $CuCl_2$ sublime at 1225.40 K and 788.48 K, respectively; at 10 Pa, the sublimation temperatures of CuCl and $CuCl_2$ are down to 1094.66 K and 726.62 K, respectively. In the studied temperature and pressure range, $CuCl_2$ decomposes directly from the solid phase. The decomposition temperatures are 644.79 K, 697.10 K and 888.00 K at 10 Pa, 100 Pa and 101,325 Pa, respectively. Therefore, under the reaction conditions, the system pressure and temperature cannot be changed to obtain gaseous $CuCl_2$.

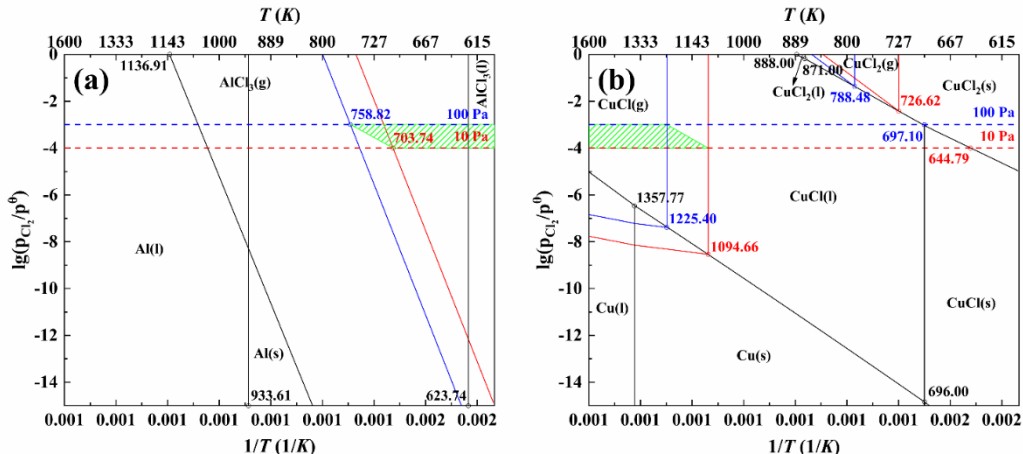

**Figure 4.** Equilibrium diagram of chloride decomposition reaction: (**a**) Al−Cl system; (**b**) Cu−Cl system.

Combining the green areas in Figure 4a,b into the same figure, we can obtain the control regions where Al−Cu chlorides are separated one by one in gaseous form. As shown in Figure 5, the green and orange areas are the control areas for the gaseous separation of $AlCl_3$ and CuCl, respectively. The reaction conditions are controlled in their respective intervals from right to left, and the corresponding chlorides can be separated from the mixture. It should be noted that, in order to prevent $CuCl_2$ in the system from decomposing $Cl_2$, the left boundary of the $AlCl_3$ control zone corresponds to the temperature at which the decomposition reaction of $CuCl_2$ occurs. Vacuum does not directly affect the equilibrium region of solid and liquid substances but has a more profound impact on the equilibrium region of gaseous substances. Under 101,325 Pa, the $\Delta G^\theta$ of Ni, Co, Mn, Li, Al, and Cu series chloride gasification is inversely proportional to the temperature T, and $lg(p_{Cl_2}/p^\theta) \equiv 0$ in Equation (2) will not affect the value of $\Delta G^\theta$; In a vacuum environment, where $lg(p_{Cl_2}/p^\theta) < 0$, $\Delta G$ is reduced to 0 more quickly, thereby reducing the boiling point of the substance and increasing the gas phase stability zone. If the boiling point is still higher than the melting point after reduction, the liquid−phase stable region will decrease, and the solid−phase stable region will remain unchanged. If the boiling point drops below the melting point, the substance skips the melting process at the corresponding boiling point temperature and sublimates directly. At this time, the liquid phase stable area disappears, and the solid phase stable area decreases.

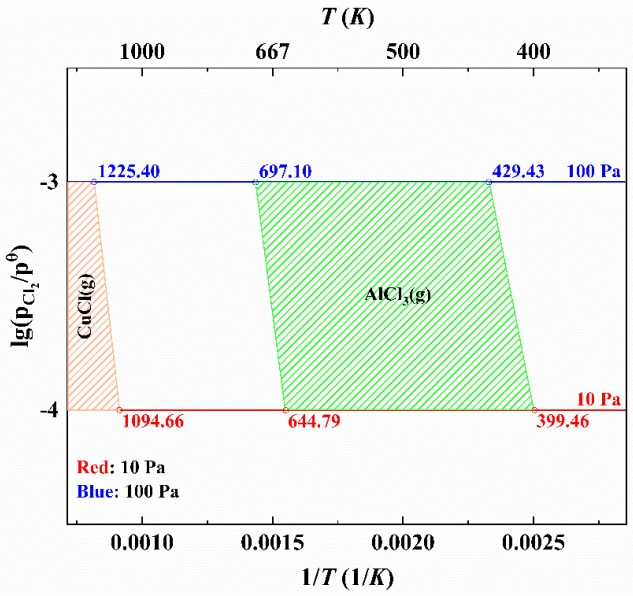

**Figure 5.** Al−Cu chloride gaseous separation control area.

It has been discussed in Figure 4b that $CuCl_2$ will not produce $CuCl_2(g)$ under the pressure of 10–100 Pa but will decompose into CuCl and $Cl_2(g)$. As shown in Figure 6, the content of each species at the equilibrium temperature at 10 Pa (Figure 6a) and the $\Delta G$ of possible chemical reactions at the corresponding temperatures (644.79 K and 1094.66 K) were discussed separately (Figure 6b). 1 mol $CuCl_2$ was put into the system, and the pressure was set to 10 Pa to ensure that all gas−phase components generated during the reaction were extracted. When the temperature is lower than 644.79 K, the decomposition temperature of $CuCl_2$ is not reached, and the system remains in its initial state. When the temperature reaches 644.79 K, $CuCl_2$ is decomposed into CuCl and $Cl_2(g)$ (Equation (4)), the generated $Cl_2(g)$ is extracted from the system, and $CuCl_2$ is almost completely converted into CuCl. Although the sublimation temperature of $CuCl_2$ at 10 Pa (726.62 K) has not been achieved at this time, the sublimation of $CuCl_2$ still occurs (Equation (5)), so the total amount of CuCl decomposed is slightly less than 1 mol. When the temperature was elevated to 1094.66 K, CuCl(l) boiled and entered the gas phase (Equation (6)). At the same time, since a large amount of CuCl(g) is contained in the gas phase at this time, a small

part of CuCl(g) undergoes a disproportionation reaction before leaving the system to form $CuCl_2(g)$ and Cu (Equation (7)).

$$CuCl_2 = CuCl_2(g) \tag{4}$$

$$2CuCl_2 = 2CuCl + Cl_2(g) \tag{5}$$

$$CuCl = CuCl(g) \tag{6}$$

$$2CuCl(g) = CuCl_2(g) + Cu \tag{7}$$

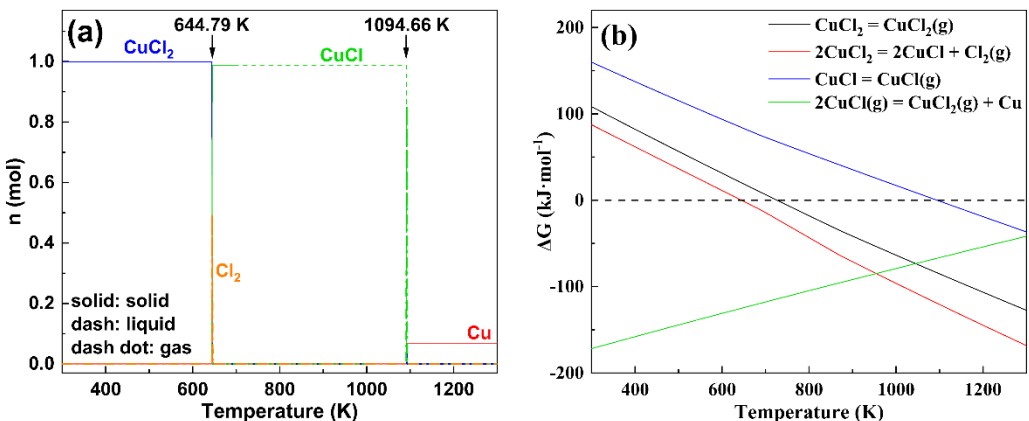

**Figure 6.** (**a**) 10 Pa, the content of each species at equilibrium temperature (**b**) $\Delta G-T$ diagram of possible chemical reactions during the vacuum decomposition of $CuCl_2$ system.

## 4. Conclusions

In this study, based on the thermodynamic theory of chlorination reactions, the feasibility of chlorinating the valuable metals Ni, Co, Mn, Li, Al, and Cu in spent lithium−ion battery systems was analyzed from a computational point of view, and the vacuum volatilization separation conditions for each chlorination product were deduced. Without introducing additional impurities, $CuCl_2$ has good chlorination capacity and can be used as a suitable chlorinating agent. Keeping the system pressure at 10 Pa, $CoCl_2$, $NiCl_2$, $MnCl_2$ and LiCl in the gas phase were obtained when temperatures reached 852.13 K, 866.15 K, 888.29 K and 996.78 K, respectively. The impurity element Al in the reaction system escapes from the system as $AlCl_3$ at 399.46 K and does not affect the subsequent vacuum volatilization of the valuable metal; the impurity element Cu is mainly converted to CuCl, which is stable under vacuum conditions and does not volatilize into the gas phase until 1094.66 K.

The process enables the chlorination and vacuum volatilization of valuable metals from lithium−ion battery cathode scrap and can be combined with a pyrotechnic pretreatment process for the chlorination and removal of impurity elements. The process helps to shorten the existing recycling process of used lithium−ion batteries, making it possible to recover and separate the valuable metals Ni, Co, Mn, and Li in a single step. At the same time, the chlorination separation process under vacuum conditions provides a solution for the separation and enrichment of valuable metals from low−grade hard−to−enrich minerals.

**Author Contributions:** Conceptualization, W.L. and G.H.; methodology, W.L., J.D. and J.W.; validation, W.L., G.H. and J.D.; formal analysis, W.L. and G.H.; resources, J.D. and J.W.; funding acquisition, J.W. and W.M.; writing—original draft preparation, W.L.; writing—review and editing, W.L. All authors have read and agreed to the published version of the manuscript.

**Funding:** This work was supported by the Candidate Talent Training Program of Yunnan Province (grant number 202005AC160041), Kunming University of Science and Technology Student Extracurricular Academic Science and Technology Innovation Fund (grant number 2022KJ194 and 2022ZK106).

**Institutional Review Board Statement:** Not applicable.

**Informed Consent Statement:** Not applicable.

**Data Availability Statement:** Not applicable.

**Conflicts of Interest:** The authors declare no conflict of interest.

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
