# Peer review of "Thermodynamics of Vacuum Chloride Volatilization of Ni, Co, Mn, Li, Al, and Cu in Spent Lithium−Ion Battery"

_metals, doi:10.3390/met12122183_

Round 1
Reviewer 1 Report (New Reviewer)
The article is devoted to the study of the processes of vacuum evaporation of Ni, Co, Mn, Li, Al and Cu chlorides in cathode materials, as well as an analysis of the processes associated with the chlorination of valuable metals. In general, the proposed direction is quite promising and has several significant significant results, however, there are a number of significant serious comments that the authors need to eliminate before the article can be considered for publication.
1. The abstract describes the observed changes in sufficient detail and is simply a listing of the facts about the studies carried out. It reflects neither the idea nor the novelty or relevance of this study, which makes it difficult to objectively evaluate this work from the very beginning. The authors are invited to give more details to this description, and also to pay attention to this remark when correcting the article.
2. Regarding formulas 1 and 2, the authors should give more information about their applicability, as well as specify formula 1 and its significance for this work.
3. The graphs presented in Figure 1, ΔG-T dependences, the authors should carefully check all the established data, as well as the values ​​of the pressures given, if there are no typos in the data presented, then the authors should give explanations for such a strong difference in pressure values.
4. The authors showed that the best results were achieved for copper and its chlorides, but the arguments presented are not enough. Authors should provide a comparative table of the main results.
5. The conclusion should also be supplemented, a number of existing comments regarding the novelty and relevance of the research topic should be corrected.
Author Response
Response to Reviewer 1 Comments
Point 1: The abstract describes the observed changes in sufficient detail and is simply a listing of the facts about the studies carried out. It reflects neither the idea nor the novelty or relevance of this study, which makes it difficult to objectively evaluate this work from the very beginning. The authors are invited to give more details to this description, and also to pay attention to this remark when correcting the article.
Response 1: Thank you for your suggestion and we have amended the abstract accordingly.
Point 2: Regarding formulas 1 and 2, the authors should give more information about their applicability, as well as specify formula 1 and its significance for this work.
Response 2: The core of the research in this paper is the removal of chlorides from the product in gaseous form under vacuum conditions. This is done through the coupling of air pressure and temperature. Regardless of the state of the chlorides before they enter the gaseous phase, we need to find out theoretically the exact temperature at which they sublimate (vaporise) for subsequent studies. It can be found that at 10 Pa the boiling points of all the chlorides listed, except LiCl and CuCl, are reduced below the melting point. This property helps to reduce the interaction of the chlorides in their molten state and facilitates the separation between them.
Point 3: The graphs presented in Figure 1, ΔG-T dependences, the authors should carefully check all the established data, as well as the values ​​of the pressures given, if there are no typos in the data presented, then the authors should give explanations for such a strong difference in pressure values.
Response 3: After careful examination, the data presented in Figure 1 were not problematic. The purpose of comparing the ΔG-T curves for each chloride at 101325 Pa (one standard atmosphere) and 10 Pa is to show that the vacuum state is conducive to achieving the volatilisation of the chloride. In other words, it is theoretically more favourable to achieve the volatilisation and purification of chloride under vacuum.
Point 4: The authors showed that the best results were achieved for copper and its chlorides, but the arguments presented are not enough. Authors should provide a comparative table of the main results.
Response 4: As can be seen from Figure 1, the ΔG-T curve for the formation of CuCl2 from Cu and Cl2 is at the top of the reactions studied, indicating that Cu has the weakest binding capacity to Cl2 and is able to achieve the chlorination process of other metals. Also, the presence of Cu as a reaction feedstock and the use of CuCl2 as a chlorinating agent avoid the introduction of other metal impurities.
Point 5: The conclusion should also be supplemented, a number of existing comments regarding the novelty and relevance of the research topic should be corrected.
Response 5: Thank you for your suggestion. We have re-examined the existing comments on the novelty and relevance of the research topic.

Reviewer 2 Report (New Reviewer)
Thermodynamics of vacuum chloride volatilization of Ni, Co, Mn, Li, Al and Cu in spent lithium-ion battery is very important paper. Some improvement is required.
Line 11: A chlorination method of the valuable metals Ni, Co, Mn and Li in the carbothermal reduction products was proposed (in which temperature interval)
Line 81: Melting and boiling points (sublimation points) of Ni, Co, Mn, Li, Al, and Cu corresponding chlorides at 101325 Pa and 10 Pa. Iron is present with Ni, Co, Mn, Li, Al and Cu in black mass of Li-ion batteries. Why did you not include Iron(II) chloride and Iron (III) chloride in this list of studied elements?
Line 137: and corresponding metals (such as…) will not be decomposed,
Line 225: This study provides an idea for realizing the separation of valuable metals Ni, Co, Mn and Li alloys in vacuum carbothermic reduction products from the perspective of pyrometallurgy. This is correct, but you did not study carbothermic reduction, only selective vacuum chlorination. Carbothermic reduction produces very dangerous compound based on carbon and chlorine. Maybe an usage of hydrogen as a reducing agent can be environmental friendly?
Line 232: Continuing to heat up to 1094.66 K, further separation of the chlorinated product Cu and the decomposition product CuCl can be achieved. In which reactor (furnace) can be selective collection of metallic chloride achieved?
Line 234: This process can be combined with pyrolytic reduction processes such as carbothermal reduction to further shorten the reaction process, make it possible to recover and separate valuable metals Ni, Co, Mn and Li by one-step method, and provide theoretical support for the subsequent treatment of related wastes. Unfortunately, no validation of theoretical results in experimental conditions. Regarding the presence of carbon together with metals in black mass, it is open question how to use this Thermodynamics of vacuum chloride volatilization of Ni, Co, Mn, Li, Al and Cu in spent lithium-ion battery for selective metal recovery.
Author Response
Response to Reviewer 2 Comments
Point 1: Line 11: A chlorination method of the valuable metals Ni, Co, Mn and Li in the carbothermal reduction products was proposed (in which temperature interval)
Response 1: Thank you for your suggestion and we have amended this statement.
Point 2: Line 81: Melting and boiling points (sublimation points) of Ni, Co, Mn, Li, Al, and Cu corresponding chlorides at 101325 Pa and 10 Pa. Iron is present with Ni, Co, Mn, Li, Al and Cu in black mass of Li-ion batteries. Why did you not include Iron(II) chloride and Iron (III) chloride in this list of studied elements?
Response 2: The lithium-ion battery cathode scrap used in this study contains almost no Fe and its compounds, so the effect of Fe is not considered in the thermodynamic calculations.
Point 3: Line 137: and corresponding metals (such as…) will not be decomposed,
Response 3: Thank you for your suggestion and we have amended this statement.
Point 4: Line 225: This study provides an idea for realizing the separation of valuable metals Ni, Co, Mn and Li alloys in vacuum carbothermic reduction products from the perspective of pyrometallurgy. This is correct, but you did not study carbothermic reduction, only selective vacuum chlorination. Carbothermic reduction produces very dangerous compound based on carbon and chlorine. Maybe an usage of hydrogen as a reducing agent can be environmental friendly?
Response 4: Thank you for your suggestion and we have amended this statement.
Point 5: Line 232: Continuing to heat up to 1094.66 K, further separation of the chlorinated product Cu and the decomposition product CuCl can be achieved. In which reactor (furnace) can be selective collection of metallic chloride achieved?
Response 5: Thank you for your suggestion and we have amended this statement.
Point 6: Line 234: This process can be combined with pyrolytic reduction processes such as carbothermal reduction to further shorten the reaction process, make it possible to recover and separate valuable metals Ni, Co, Mn and Li by one-step method, and provide theoretical support for the subsequent treatment of related wastes. Unfortunately, no validation of theoretical results in experimental conditions. Regarding the presence of carbon together with metals in black mass, it is open question how to use this Thermodynamics of vacuum chloride volatilization of Ni, Co, Mn, Li, Al and Cu in spent lithium-ion battery for selective metal recovery.
Response 6: The focus of this study is to analyse the feasibility of chloride chlorination of the valuable metals Ni, Co, Mn, Li, Al and Cu in spent lithium-ion battery systems from a computational perspective and to deduce the conditions for the vacuum volatilisation separation of each chlorination product. The relevant experiments are subject to further study.

Reviewer 3 Report (New Reviewer)
Dear Authors,
this paper indicated good topic to investigate, chloride shows better bolatarization. it can be published after following correction.
Fig.1 shows kJ/mol and Fig.6 shows kJ mol-1. it should be consistent.
Fig.1 shows standard gibbs energy and Fig.6 shows ΔG. it should be consistent.
Fig. 1a shows standard gibbs energy and Fig.1b shows gibbs energy. is it because a is standard pressure? but temperature is not same. but if you are indicate intentionally, it is ok.
References
1. is bold
some name is initial then last name order, and other name is last name then initial. it should be consistent.
Author Response
Response to Reviewer 3 Comments
Point 1: Fig.1 shows kJ/mol and Fig.6 shows kJ mol-1. it should be consistent.
Response 1: Thank you for your suggestion. The physical quantities and units of the vertical axis were incorrectly labelled in Figure 6(a) of the original manuscript and have now been corrected.
Point 2: Fig.1 shows standard gibbs energy and Fig.6 shows ΔG. it should be consistent.
Response 2: The results of thermodynamic calculations at the standard state (one atmosphere) are marked as ΔGÆŸ and those at the non-standard state (10 Pa) are marked as ΔG.
Point 3: Fig. 1a shows standard gibbs energy and Fig.1b shows gibbs energy. is it because a is standard pressure? but temperature is not same. but if you are indicate intentionally, it is ok.
Response 3: Yes, ΔGÆŸ and ΔG are in two different states, so there is a distinction between a standard state and a non-standard state.
Point 4: References
- is bold
some name is initial then last name order, and other name is last name then initial. it should be consistent.
Response 4: Thank you for your suggestion. We have re-examined the citation format of the references and corrected errors.

Round 2
Reviewer 1 Report (New Reviewer)
The authors answered all the questions, the article can be accepted for publication.
This manuscript is a resubmission of an earlier submission. The following is a list of the peer review reports and author responses from that submission.
Round 1
Reviewer 1 Report
Manuscript number: #1856147
Manuscript title: Thermodynamics of vacuum chloride volatilization of Ni, Co, Mn, Li, Al and Cu in spent lithium-ion battery
Authors: Junshuai Ding, Guochen Hu, Wen Luo, Jijun Wu, Wenhui Ma
General Impression##1856147:
The presented manuscript deals with a globally important topic – litium-ion battery waste management. The paper gathers several scientific disciplines: environmental, metallurgical and chemical engineering. This research also indicates the contribution to the concept of circular economy (actually, this was not accented in the paper, but may be a significant contribution – please, reconsider this as a reviewer’s suggestion). Resource recovery and waste minimization are of special concern in circular economy (reflecting to sustainable waste management), and these aspects certainly have to be accented through the manuscript.
General comments#973509:
Please avoid the use of abbreviations in the Abstract (‘NCM’ cathode); initially, each abbreviation has to be introduced when first time mentioned in the text (of the manuscript).
It is strongly recommended that English should be proof read by native English person, or competent professionals.
The manuscript is well structured and decently written (text flow is easy to follow).
The results are clearly presented, but not strongly compared with those in the literature.
The conclusions are formulated concisely, but not being profoundly supported by the previous research.
The key messages/outcomes of the manuscript are clear and concise.
The references are well selected (up-to-dated) and punctually cited; the number of relevant publications may be increased (15 references seems not to be convincible enough as the state of the art).
For these reasons, it is the reviewer's strong recommendation that manuscript will be accepted for publication in the Metals journal after addressing following reviewer’s suggestions.
Reviewer 2 Report
The manuscript presents an idea of a novel pyrometallurgical method to separate metals from waste lithium-ion batteries. The manuscript is well written, the presented idea is interesting and potential and the societal impact is significant (circular economy and electrification of societies).
However, the content is way too thin for a full research paper and the scientific content is also very shallow. The entire manuscript is based on thermodynamic calculations, but the calculations are just classical thermodynamics and authors do not make any own contribution to the development and/or modification of the models. When this is the case the suggested process would be definitely needed to be studied experimentally also. I.e. the work here is a good theoretical basis, but only basis, for a study that presents a novel process idea based on experimental work.
Based on the above reasons, my clear recommendation is to reject the manuscript.
Reviewer 3 Report
Dear authors
Manuscript deals about an interesting topic, the recovery of metals from end life batteries. However, I consider that manuscript needs to be improved. The idea of the research is unclear, the Introduction needs to be improved with a review of technologies studied for the recovery of metals from this waste. A wide thermodynamic description of involved reactions needs to be performed. Conclusions needs to be improved as they need to be obtained from research performed by authors.
For example, authors said that "oxalate and carbonate in the precipitation of valuable metal compounds will release a large amount of CO2 in the subsequent battery manufacturing cycle process, which is not conducive to the realization of carbon emission reduction targets such as "carbon peak" and "carbon neutrality". However, the pyrometallurgical route is related to higher CO2 emissions.
Can be classified the chlorination process as a green method? Temperatures used are high and chloride processes are expensive.
Introduction needs to be improved with a wide description of pyrometallurgical and hydrometallurgical processes used for the recovery of metales from these wastes and to highlight the novelty and need of this research
Materials and method. Authors have used any software to perform the calculations?
Conclusions: 852.13 K, 866.15 K, 888.29 K are very similar. Do the authors believe that it would be possible to achieve the separation industrially? Additionally, the different kinetics can make separation difficult.
Conclusions needs to be obtained from results perform by authors. For example "The technological process proposed in this study is relatively simple, and achieves the dual purpose of reducing energy consumption and cost." Authors do not perform any comparison with other processes.
Why can be combined with pyrolysis? Carbothermal reduction is a pyrolysis process?
Obtaining thermodynamic data is not clear. The authors only refer to sublimation process, how was performed the chlorination of different metals? Equation 3 refers to Cl2 not to CuCl2. What temperatures and conditions needs to be used? For some metals is higher than sublimation temperatures?